# Improvements on Live Feed Enrichments for Pikeperch (*Sander lucioperca*) Larval Culture

**DOI:** 10.3390/ani10030401

**Published:** 2020-02-29

**Authors:** Carlos Yanes-Roca, Astrid Holzer, Jan Mraz, Lukas Veselý, Oleksandr Malinovskyi, Tomas Policar

**Affiliations:** 1South Bohemian Research Center of Aquaculture and Biodiversity of Hydrocenoses, Faculty of Fisheries and Protection of Waters, University of South Bohemia in Ceske Budejovice, Zátiší 728, 389 25 Vodňany, Czech Republic; jmraz@frov.jcu.cz (J.M.); veselyl@frov.jcu.cz (L.V.); omalinovskyi@frov.jcu.cz (O.M.); policar@frov.jcu.cz (T.P.); 2Institute of Parasitology, Biology Centre of the Czech Academy of Sciences, Branišovská 31, 370 05 České Budéjovice, Czech Republic; astrid.holzer@paru.cas.cz

**Keywords:** Chlorella, rotifers, first feeding, fatty acids, *Sander lucioperca* larvae

## Abstract

**Simple Summary:**

Pikeperch (*Sander lucioperca*) is considered a species of high interest for the development of new species in the European Union. Currently, survival rates during the larval stages are below 20%. Inadequate larval rearing protocols, such as poor nutrition, are responsible for such low survival, which is stopping the commercial development of pikeperch. In order to improve and customize nutritional needs during the larval stages, the use of Chlorella vulgaris was introduced in the enrichment feeding protocol. The introduction of such algae to their diet through rotifers has improved survival and overall fitness by providing adequate fatty acids to their nutrition.

**Abstract:**

This study focused on supplementing pikeperch (*Sander lucioperca*) larvae with rotifers fed with *Chlorella vulgaris* during the first 15 days post hatching (dph). Larvae were fed a combination of rotifers and artemia under three different enrichments: A) *Nannochloropsis occulata*, B) *Chlorella vulgaris*, and C) a commercial enrichment—Selco, Spresso from INVE. After 17 days from the trial initiation differences were found between treatments on survival rate, myomere height (MH), fatty acid composition, and stress tolerance. In terms of survival, larvae from treatment b (74.5%) and c (66%) excelled over the control (a) treatment (59%). Furthermore, larvae from both the Chlorella (b) and the Selco (c) treatments showed more resilience to stress conditions (10% and 37% reduction in mortality) when exposed to high salinity conditions (18ppt) for 3 h (stress response). Overall, larvae from treatments b and c performed better than those receiving a non-enriched diet (a), likely due to the higher levels of Essential Fatty Acids (EFA) and the ability of pikeperch to desaturate and elongate fatty acids (FA) with 18 carbons to LC PUFAs (Polyunsaturated Fatty Acids). The present study provides valuable input for designing improved feeding protocols, which will increase the efficiency of pikeperch larval culture.

## 1. Introduction

Pikeperch (*Sander lucioperca*) has been chosen by several international programs looking for aquaculture diversification within Europe [1]. This fresh and brackish water species, commonly found in Central, Eastern, and Northern Europe, [2], is highly demanded by the gastronomic industry and the recreational angling community [3]. Most of pikeperch production currently comes from wild fisheries, but production in Recirculating Aquaculture Systems (RAS) is increasing (FAO, 2013), due to its high market value and fast growth rate in RAS [4,5,6,7].

Despite good knowledge of the nutritional requirements of juvenile pikeperch [8,9,10], larval culture remains a bottleneck. Current research is focused on further improving the low effectiveness and high costs of rearing larval pikeperch in RAS. Mass production of pikeperch depends on the development of culture techniques in RAS so that sufficient quantities of juveniles can be produced [11,12].

The introduction of rotifers (*Brachionus plicatilis*), to pikeperch larval culture has been successful, improving survival and overall fitness rates similar to those in marine species of economic value, such as grey mullet (*Mugil cephalus*) [13], sole (*Solea solea*) [14,15], gilthead seabream (*Sparus aurata*) [16,17], and sea bass (*Dicentrarchus labrax*) [18].

One of rotifers’ main characteristics is their ability to absorb and retain the nutritional composition of any diet that they are exposed to. They are, hence, regarded as living food capsules for transferring nutrients to fish larvae. These nutrients include highly unsaturated fatty acids (mainly 20:5n-3 and 22:6n-3) essential for the survival of marine fish larvae [19], as well as pikeperch [20]. Microalgae are one of the most common feeds given to rotifers [11] to enhance their nutritional composition for fish.

In the last half-century, several hundred microalgae species have been tested as feed for live feed, but only about twenty species have gained widespread use in aquaculture, with the most popular ones being *Isochrysis* spp., *Pavlova lutheri*, *Tetraselmis suecica*, *Nannochloropsis* spp. and *Chaetoceros* spp. [21]. Suitable candidates for use in aquaculture must have rapid growth rates, be easy to culture in large-scale facilities, and have a good nutrient composition [21]. Evaluation of the nutritional value of microalgae focuses on certain biochemical constituents, predominantly fatty acids, especially polyunsaturated fatty acids (PUFA), vitamins and amino acids [22]. The PUFA profile of microalgae varies significantly among taxonomic groups and can also be controlled, at least partly, through manipulating the growth conditions of the microalgae [23].

Since the late 1980s, a condensed freshwater *Chlorella* sp. has been widely used for the production of the rotifer *Brachionus* spp. [24] in Japan, mainly as “green water”, due to its anti-bacterial characteristics and nutritional value.

*Chlorella vulgaris* is a fast growing unicellular green algae and has been widely used as a human food supplement [25], and was found to be rich in n-3 long-chain polyunsaturated fatty acids (LC-PUFA). This green algae has been incorporated into fish diets, and has been fed to Ayu (*Plecoglossus altivelis*) and the Korean rockfish (*Sebastes schlegeli*) [26,27]. It has been noted in previous studies that inclusion of 2.5–10% of algae into fish diets enhanced growth performance, feed utilization efficiency, and physiological activity [28]. However, *Chlorella* has not previously been tested as a feed ingredient for pikeperch.

The aim of this study was to enhance pikeperch nutrition by providing larvae with a diet adjusted to pikeperch’s fatty acids metabolism, during the first 21 days post hatching.

## 2. Materials and Methods

The trial was run at the University of South Bohemia, Facility of Fisheries and Protection of Waters, Czech Republic (USB, FFPW). Spawning and fertilized egg production were obtained from pond-cultured pikeperch broodstock [29,30] (TL = 517 ± 35 mm and W = 1215 ± 200 g), held at the same facility under controlled conditions [31] in RAS. Used broodstock was regularly spawned since 2017, Final oocyte and sperm maturation was performed under spawned 15:9 h light:darkness regime with a light intensity of 100 lux, water temperature of 15 ± 0.5 °C [6,31,32], and synchronized with a dose of the intramuscular hormonal injection Human Chorionic Gonadotropin (hCG; Chorulon, Intervet International B.V.) at 500 IU.kg^−1^, as previously cited by Křištan [33] and Blecha [31]. All broodstock were anesthetized with clove oil (Dr. Kulich Pharma Ltd., Hradec Králové, Czech Republic) at a concentration 30 mgL^−1^ [34], before manipulation. After hormonal treatment, pairs composed of both sexes were separated and stocked in RAS tanks for nest spawning, following the research method set forth by Malinovskyi [29,30]. After spawning, egg fertilization, and laying, broodstock were removed and eggs on the nest were incubated in each tank under a water temperature of 16 ± 0.5 °C for 8 days until hatching occurred [30]. Three day old larvae from one female (100 per L) were stocked into 2 L larval rearing tanks (n = 12). Water quality parameters in the RAS were monitored daily; average values were salinity (3 ± 0.5 ppt), dissolved oxygen (8.0 ± 1 mgL^−l^), temperature (17.1 ± 0.2 °C). Ammonia, Nitrite and nitrate levels were measured every three days. Ammonia (NH_3_ = 0.21 ± 0.05 mgL^−1^), nitrite (NO_2_ = 0.02 ± 0.01 mgL^−1^) and nitrate (NO_3_ = 10 ± 0.2 ppm). Larvae were exposed to a 13L/9D photoperiod and to dim light conditions (300 LUX) during the light hours. Daily siphoning was carried out to remove any debris from the bottom of the tanks. In order to reduce the lipid layer, surface skimmer we present in each tank.

Three treatments were tested. The first was control treatment (A), where larvae were offered rotifers (*Brachionus plicatilis*) fed with *Nannochloropsis occulata* during the first eleven days (15 dph), and unenriched artemia until the end of the trial. Treatment B followed the same feeding protocol as treatment A, but rotifers (*Brachionus plicatilis*) supplied to the larvae were fed with *Chlorella vulgaris*. *Artemia salina* provided to larvae from treatment B was not enriched, in order to assess the potential long-term effects of the *Chlorella vulgaris* supplementation on the larvae fatty acids’ composition and survival. The third treatment (C) used rotifers (fed on *Nannoclopropsis*) and artemia, both enriched with Spresso emulsion by Selco (INVE, Salt Lake City, UT, USA).

Rotifers were fed to the larvae three times per day (08:00, 11:30 and 15:30 h) starting at 4 dph, until 15 dph, with an initial concentration of 10 individuals per ml. Rotifers’ density was increased to 14 per ml from day 8 post hatching until day 12 post hatching. From day 13 post hatching, rotifers’ density was progressively reduced, offering 10 rotifers per ml at day 13 and 8 rot/mL at day 14 post hatching. No more rotifers were added to the system after day 15 post hatching. Artemia feeding was applied in each experimental group from day 12 post hatching at a density of 2 artemia per ml. Prior to each feeding, residual counts were measured and feeding densities were steadily increased based on the counts. At day 13 and 14 post hatching, density was increased to 3 and 4 per ml, respectively. On day 15 and 16 post hatching, artemia density was 7 per ml, and from day 17 to the end of the trial artemia density was 8 per ml. By 21 dph, rotifer density was 0 rotifers mL^−1^ and 8 artemia mL^−1^. Live feed culture for the trial was done onsite. Rotifers (average size of 280 µm) were produced following a batch culture protocol.

Rotifers used for treatment A were fed with *N. occulata* (Nanno 3600, Reed Mariculture, Campbell, CA, USA) at a rate of 1 mL of paste per liter of culture twice a day. Rotifers used on treatment B were fed live *Chlorella vulgaris*, provided by Algatech (Trebon, Czech Republic). Rotifers for treatment C were fed *N. occulata* (Nanno 3600, Reed Mariculture, Campbell, CA, USA), and enriched 12 h prior to feeding with Selco Spresso (INVE). Artemia nauplii (Micro Artemia cysts, Ocean Nutrition^TM^, Belgium) were hatched (12 h) onsite and fed right away without any enrichment to treatments A and B. On the other hand, artemia for the treatment C, was enriched with Selco Spresso (INVE) 12–14 h prior to feeding. Artemia nauplii’s average size was 430 µm.

RAS flow rates started at 100 mL.min^−1^ and increased with time; by the end of the trial, flow rate was 300 mL.min^−1^. Prior to each feeding, flow was stopped and re-started 2 h after, in order to improve larval feeding efficiency.

A pooled sample of one hundred 3 dph larvae was collected to record their total length (TL), myomere height (MH), and eye diameter (ED). Seven and eleven days after treatment initiation (12 and 16 dph), a pooled sample of forty larvae per treatment (10 per tank) were collected using a 300-micron-diameter mesh. Their TL, MH, ED, and stomach fullness (SF), were recorded using an Olympus (Tokyo, Japan) BX41 microscope fitted with a Canon-72 (Tokyo, Japan) digital camera and Olympus (Tokyo, Japan) cellSens imaging software (version 1.3).

Prior to the appearance of cannibalism and light photosensitivity, the trial was terminated at twenty-one dph. All larvae were accounted for and samples were collected. A pooled sample of sixty larvae per treatment were collected for FA analysis at day 12, day 21, shock frozen, and stored at −80 °C. The diets (prey organisms) themselves were also analyzed (3 mg) for FA composition. Thirty larvae per treatment were fixed in RNA later^@^ to determine RNA–DNA ratios. After penetration of the larvae with the RNA preservative for 3 h at room temperature, they were transferred to −80 °C freezer for storage. Another 100 larvae per treatment were collected for final morphometric analysis (TL, MH, ED, SF).

### 2.1. Fatty Acid Analysis

All frozen samples were analyzed at the USB, FFPW, Laboratory of Nutrition. Lipid extraction was carried out following the protocol of Hara and Radin (1978) with slight modifications. In brief, approximately 0.05 g of larvae samples were added to 1 mL of deionized water and the mixture was homogenized in 10 mL of hexane–isopropanol (3:2) and 6 mL of Na_2_SO_4_ (6.67%) were added to the obtained homogenates and mixed. After centrifugation, the upper lipid phase was transferred into pre-weighted tubes and subsequently evaporated under nitrogen. Final determination of lipid content was carried out gravimetrically.

Methylation of 1 mg of lipids was induced with boron trifluoride–methanol complex solution and NaOH as described by Appelqvist, (1968). Resulting fatty acid methyl esters (FAME) were checked on a Thin Layer Chromatography (TLC) plate and analyzed using a gas chromatograph (Trace Ultra FID; Thermo Scientific, Thermo Fisher Scientific, Waltham, MA, USA) equipped with a BPX 70 column (SGE, Raleigh, NC, USA). Subsequently, a comparison of FAME retention times for the sample and standards GLC-68D was used to identify individual fatty acids.

The methods used for lipid extraction and methylation of rotifers and artemia followed the same protocol as the larval analysis [35,36].

### 2.2. RNA/DNA Ratio Analysis Method

For RNA/DNA ratio analysis, frozen larvae were completely defrosted and picked from the eppendorfs using sterile forceps. DNA and RNA was extracted individually from six to eight individual larvae (per diet), using the All Prep RNA/DNA Mini Kit (Qiagen, Thermo Fisher technology, Waltham, MA, USA). Concentrations, quality and purity (260/280 and 260/230 ratios) of DNA and RNA were determined by nanodrop.

### 2.3. Salinity Stress Challenge

Twenty-one days after hatching, 100 larvae per treatment (25 per tank) were collected and transferred to a 2 L tank (n = 3), where they were exposed to a salinity of 18 ppt for three hours. Larval mortality was recorded in each tank every ten minutes during the first hour, then recorded at 120, 130, 140, 150 and 180 min. from the initial stocking. Water quality conditions were kept the same as the original trial tanks, with the exception of salinity (18 ppt).

Larvae during this trial were handled in accordance with national and international guidelines for the protection of animal welfare (EU-harmonized Animal Welfare Act of the Czech Republic). The experimental unit is licensed (No. 2293/2015-MZE-17214 and No. 55187/2016-MZE-17214 in project NAZV QK1820354) according to the Czech National Directive (Law against Animal Cruelty, No. 246/1992).

### 2.4. Statistical Analysis

Differences in body measurements, food consumption, and FA composition between the three different enrichments in larvae were sampled at 12, 16 and 21dph and evaluated with linear mixed models (LMM, package *lme4*, version 1.1-7; [37]). The effect of the enrichment was tested on fish total length, MH, and ED (response variables), and the tank was included as a random effect. Prior to LMM, the different response variables were transformed with the Box–Cox transformation, which gives the best power estimate for each variable (package *car*, version 2.1.2; [38]). Thereafter, multiple pairwise comparisons between enrichments were obtained using Tukey’s all-pair comparisons, applying the Bonferroni correction to adjust the p-values (package *multcomp*, version 1.3-3; [39]). The same analyses were run to test for differences in the fatty acid composition between enrichments and between artemia and rotifers used as preys (Linoleic Acid (LA), Alpha linoleic acid (ALA), Arachidonic acid (ARA), Eicosapentanoic acid (EPA) and Docosahexaenoic acid (DHA) as different response variables).

Differences in stomach fullness were assessed using the method described by Tielmann [40] (1 to 4, 1 being empty gut and 4 being full gut). Data were evaluated with generalized linear mixed models (GLMM, package *lme4*), fitted with a binomial error structure, and stomach fullness was used as a response variable. The tank was a random factor. These analyses were followed by multiple pairwise comparisons with Tukey’s all-pair comparisons.

The survival of pikeperch fish was compared between enrichment groups, using a Generalized Linear Mixed Model (GLMM). The survival rate was observed (i.e., the proportion of alive fish at 21dph as a response variable), fitted with a binomial error structure, and with enrichment as a fixed effect. The tank was a random effect. After GLMM, pairwise comparisons were obtained with Tukey’s all-pair comparison test. Bonferroni correction was applied to adjust the p-values of multiple comparisons.

RNA/DNA ratios, transformed with Box–Cox transformation, were compared between enrichments by a Linear Mixed Model (LMM), with a ratio as response variable and enrichments as a random effect, followed by Tukey’s all-pair comparison test to obtain pairwise comparisons between enrichments. Bonferroni correction was applied to adjust the p-values of multiple comparisons.

To test the salinity stress tolerance response among the treatments, a non-parametric survival analysis (Kaplan–Meier method) was performed for all groups, using survival package [41].

All analyses were conducted in R (R Core Team, Vienna, Austria [42]) and statistical significance was set at α = 0.05.

## 3. Results

### 3.1. Survival

Survival rates were significantly different between all enrichments (GLMM and pairwise comparisons *p* < 0.001), showing that the survival of larvae fed exclusively with rotifers enriched with *Chlorella* (B) was 1.44 times higher than larvae fed with *Nannochloropsis* (A) and 1.11 times higher than larvae receiving the commercial enrichment (C), while the survival of larvae from treatment C was 1.29 times higher than of larvae from treatment A (Figure 1A).

### 3.2. Larval Growth

Initial pikeperch larval total length at 3 dph was 5.25 ± 0.5 mm. After 12 days, treatment B (Figure 2A) had the larvae with the largest average total length (6.46 ± 0.68 mm). By the end of the trial (21 dph), average total length was greater in treatment B (10.91 ± 1.35 mm) than in treatments A and C (Figure 2A), but no significant treatment differences (LMM, *p*-value > 0.05) were found in total length at any sampling point.

A similar pattern was found in myomere height (Figure 2B), where no significant differences were detected (LMM, *p*-value > 0.05), with an exception on day 21, where a treatment effect was found (LMM *p*-value < 0.001). When looking at the eye diameter, no significant differences were found (LMM, *p*-value > 0.05). No significant differences in stomach fullness of larvae were found between treatments (GLMM *p*-value < 0.05) and all prey was ingested by larvae, regardless of the enrichment (Figure 2D).

### 3.3. RNA/DNA Ratio

The RNA/DNA ratio analysis showed no significant differences (LMM, *p*-value > 0.05) between enrichments (Figure 2C), yet a clear pattern was observed where *Nannochloropsis* (A) enrichment had a 1.11- and 1.84 times higher RNA/DNA ratio than the *Chlorella* (B) and Spresso enrichments (C), respectively (Figure 2C). Enrichment A showed the highest variability in RNA/DNA ratios. Pairwise comparisons were concordant with LMM results. No significant difference was found between treatment A and B (LMM, *p*-value > 0.05).

### 3.4. Salinity Stress Tolerance

Larvae exposed to 18 ppt salinity from the different treatments reacted differently over time (Figure 1B). Larvae from the control treatment and the B treatment experienced mortality from the beginning of the exposure, having a 10% and 7% mortality after 30 min of exposure. On the other hand, larvae from treatment C had a 3% mortality during the same period. After one hour of exposure, larvae from the control treatment had the highest mortality (52%), followed by the B treatment (44%) and treatment C (23%). Mortality in the control treatment slowed down in the following hour, resulting in an overall mortality of 47% (a 15% mortality increase). On the other hand, tanks with larvae from the C and B treatments experienced an increase in mortality (17% and 18%) during the same period of time (Figure 2B). Yet, larvae from the C treatment had a lower mortality (40%) after 2 h, compared to 53% and 70% from the control and B treatments, respectively. No larvae were alive after 150 min of exposure time in the control treatment. After 3 h of exposure, tanks from the B and C treatment had surviving larvae. Treatment C had the lowest final mortality (63%) compared to the B treatment (90%), and a significant difference (*p*-value < 0.05) was found in mortality between treatments after three hours.

### 3.5. Fatty Acids

The fatty acid composition of artemia and rotifers enriched with the different live feeds and commercial enrichment are shown in Table 1.

Rotifers fed *Chlorella* (ROT B) and the Spresso diet (ROT C) had 1.84- and 1.73 times higher LA (C18:2n-6) levels, and 3.22 and 3.01 times higher ALA (C18:3n-3), respectively, than rotifers fed on *Nannocloropsis* (ROT A). DHA (C22:6n-3) levels were 2.61 and 2.72 times higher (Table 1) in rotifers enriched with *Chlorella* (ROT B) and the commercial enrichment (ROT C) than rotifers fed *Nannocloropsis* (ROT A). However, rotifers fed *Nannochloropsis* (ROT A) had 2.41 and 2.81 times higher ARA (C20:4n-6) values and 5.94 and 6.29 times higher EPA (C20:5n-3) than those rotifers fed *Chlorella* (ROT B) (Table 1) and the ones enriched with Spresso (ROT C), respectively (LMM analyses, all with *p*-value < 0.001).

The fatty acid composition of larvae receiving the different enrichments at days 12 and 21 post hatching are shown in Table 2.

At 12 dph, LA levels were highest in larvae from treatment B, which were 1.15 and 1.17 times higher than in larvae from treatments A and C, respectively (LMM, *p*-value < 0.001). No significant differences were found between treatments A and C (Figure 3A).

ALA level was highest in treatment B (Figure 3B), which was 1.76 and 1.98 times higher than in larvae from treatments A and C, respectively (LMM, *p*-value < 0.001), and with no significant differences between these two groups (A and C) (LMM, *p*-value > 0.05). No significant difference between treatments were found in DHA levels (LMM, *p*-value < 0.001), although the values from treatments B and C were higher than larvae fed with *Nannocloropsis* (A) (Figure 3D). Similar results (Figure 3C) were found between larvae from the different treatments in their levels of EPA (LMM, *p*-value > 0.05). When looking at ARA levels (Figure 3E), values from the treatment B were 1.06 and 1.19 times higher than treatments A and C (LMM, *p*-value < 0.001).

After 21 days post hatching, larvae from treatment C had 1.21 and 1.19 times significantly higher LA values (Figure 3A) than larvae from treatments A and B (LMM, *p*-value < 0.001). No significant differences were found between larvae from the different treatments in their levels of ALA and EPA (LMM, *p*-value > 0.05). Furthermore, larvae from treatment A showed slightly higher EPA values (Figure 3C) than larvae from treatment C (LMM, *p*-value < 0.001). DHA levels (Figure 3D) were higher in larvae fed with the spresso enrichment (C), with a difference of 1.54 and 1.29 times compared to larvae fed only with *Chlorella* (B) and *Nannocloropsis* (C), respectively (LMM, *p*-value < 0.001).

Furthermore, larvae from the treatment B showed 1.19 times higher DHA values than larvae fed with *Nannochloropsis* (A), although no significant difference was found (LMM, *p*-value > 0.05). A significant difference between treatments was found regarding ARA levels (Figure 3E), where larvae from treatment A had the highest levels on day 21 post exposure. Pairwise comparisons were generally concordant with LMM results.

## 4. Discussion

Nutritional improvements often have a direct effect on larval growth, which allows them to overcome size-related problems [43,44]. Kestemont [8], highlighted the limitation in the range of food used for the cultured pikeperch, which can lead to nutritional imbalances or deficiencies. In order to tackle such limitations, a common practice in marine larviculture has been introduced, which uses rotifers and artemia as carriers to enhance the nutritional value of the prey offered. This well-established practice in marine larvae culture [45] has not been taken into consideration in pikeperch larval culture, despite positive conclusions from several studies [46,47]. Although larval growth during this experiment did not significantly improve when comparing total length, eye diameter, and RNA/DNA ratio between treatments, the results matched recent findings from Lund [47]. Castell [48], demonstrated that rainbow trout (*Oncorhynchus mykiss*) had severely depressed growth when reared on fat-free diets or diets low in the n−3 series, matching the pattern observed during this trial, where the larvae from the treatment with lower n−3 series (A) were smaller. A longer treatment exposure might have resulted in more significant differences, since growth depression is increased when EFA reserves start to be depleted.

The trial’s overall larval survival had an average of over 60% after 21 days and, in contrast to growth, significant differences between treatments were found. Treatment B had the highest survival compared to the commercial diet and the control. Such differences could lie in the positive effects that higher concentrations of LA and ALA had on the overall larval performance and quality, matching conclusions reached by other studies [49,50] on several species: red sea bream (*Pagrus major*), yellowtail (*Seriola lalandi*), striped knifejaw (*Oplegnathus fasciatus*), Japanese flounder (*Paralichthys olivaceus*), and turbot (*Scophthalmus maximus*). The results of this experiment reinforce the importance of adequate quantities of fatty acids during larval development [51,52].

Acute larval mortality after exposure to high salinity was observed in larvae from treatments A and B. Such mortality was not caused by air exposure or physical injury, since larvae from treatment C did not experience such mortality. Rapid swimming motion, followed by sudden death, was observed in the groups where no extra fatty acids were added. Similar responses to stress have been described and are thought to be due to a deficiency of EFA [47,53,54,55]. More specifically, such behavior had been frequently reported in sensitive species [56,57], such as grouper (*Epinephelus septemfasciatus*), when handled. Dhert [58], attributed such a stress response to diet nutritional deficiencies, which Sinnhuber [59], also observed in freshwater species, such as rainbow trout (*Oncorhynchus mykiss*), when lacking LA and ALA and exposed to excess stress conditions.

Twenty minutes after the initial exposure, gradual mortality over time was probably induced by salinity. The tank with larvae from treatment A had no larvae surviving after 150 min. On the other hand, the two tanks with larvae from treatment B and C had larvae surviving after 3 h (10% and 37%, respectively). Such differences can be attributed to the positive influence that higher levels of DHA have on larval stress tolerance, as described by Lund [47]. The same author suggested that low levels of DHA in the neuronal and membrane tissue might be the main cause of pikeperch sensitivity [60], due to a lack of fuel metabolism, since pikeperch is known to prefer the use of LC-PUFAs (DHA, EPA) as metabolic substrates [61].

Contrary to what Lund [47,60], stated, during the first 12 days, larvae appeared to be able to convert LA and ALA to 22:6n−3, as the concentration of these EFA in larval groups with higher concentrations of LA and ALA matched DHA levels from those larvae supplemented with extra DHA (commercial enrichment). After 15 days, treatments A and B were fed non-enriched artemia and a difference in DHA levels was observed when compared to those larvae receiving the DHA-enriched diet. Thus, once the source of high concentrations of LA and ALA (*Chlorella*) was removed from the larval diet in treatment B, DHA levels dropped. The results clearly indicate that pikeperch seems to have the ability to desaturate and elongate FAs, with 18 carbons to LC PUFAs, contrary to pike (*Esox Lucius*), which seems to have lost this ability as a consequence of their strict predation on other fish species capable of converting 18:3n−3 to the corresponding LC PUFAs [62].

## 5. Conclusions

The use of live feed enrichments to feed pikeperch larvae has shown to benefit larval development during the first 21 dph. The key factors for such improvements are due to the added nutritional value given by the diets. Rotifers fed with *Chlorella vulgaris* (B) are supplying the larvae with a much more complete diet in terms of EFA when compared to those fed with *N. occulata* (A), allowing the larvae to improve their development and increase their stress tolerance. The key difference between the diets lies in the amount of LA and ALA, which are 1.8 and 3.2 times higher in *Chlorella* (treatment B) than in the treatment receiving only *Nannochloropsis* (A), and the ability that pikeperch seems to have to desaturate and elongate FAs with 18 carbons to LC PUFAs. The results of this study therefore recommend the use of *C. vulgaris* for rotifer enrichment in pikeperch larvae feeding during the first 15 days post hatching, as well as an extra supplementation of DHA once larvae are feeding on artemia. Future work is recommended to further customize pikeperch nutritional needs during the first 21 days post hatching, such as using other microalgae with better EFA profiles and a more in-depth study of pikeperch’s ability to modify exogenous PUFA by desaturation and elongation.

## Figures and Tables

**Figure 1 animals-10-00401-f001:**
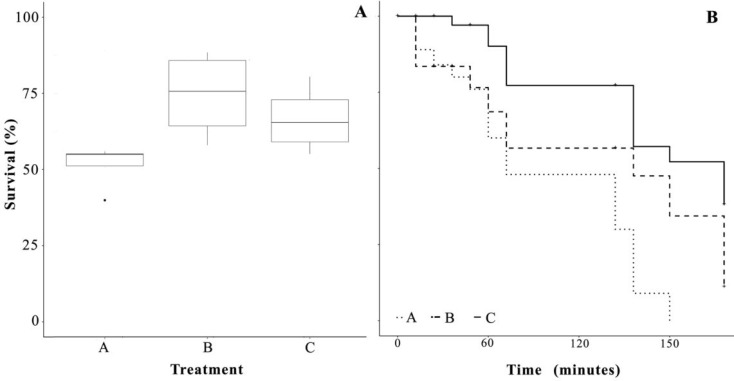
(**A**) Larval survival percentage, (**B**) Salinity stress test mortality during 3 h (n = 100), after 21 dph. Dots shown are the out layers, whiskers indicate the maximum and minimum values excluding out layers, the line in the middle of box is the median value, and upper and lower quartiles are the ends of the box. Statistically significant differences between treatments are marked with an asterisk.

**Figure 2 animals-10-00401-f002:**
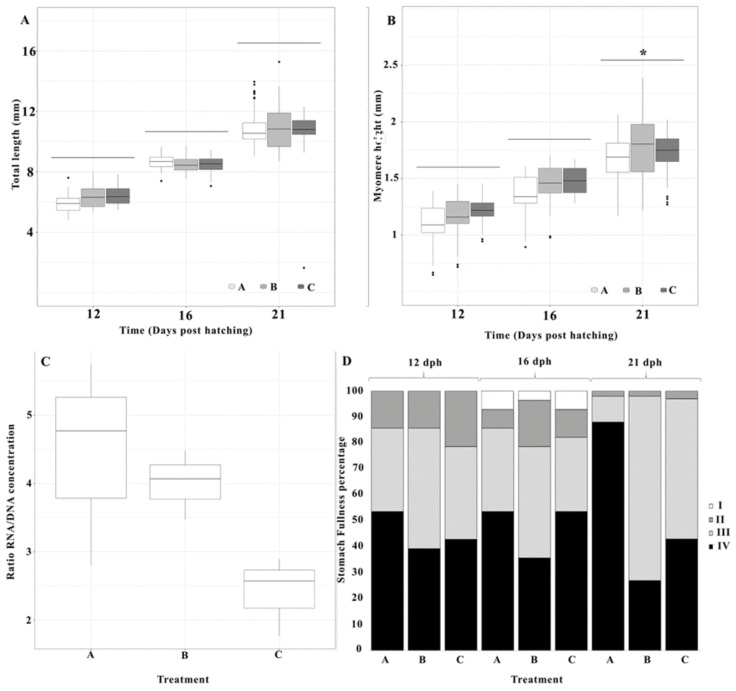
Larval growth parameters and stomach fullness from three diet treatments at days 12 (n = 40), 16 (n = 40) and 21 dph (n = 100). (**A**) Total length, (**B**) Myomere height, (**C**) RNA/DNA ratio (21 dph), (**D**) Stomach fullness expressed in percentage (1 to 4, with 4 being the maximum fullness, from darkest to lightest grey). Dots shown are the out layers, whiskers indicate the maximum and minimum values excluding out layers, the line in the middle of box is the median value and upper and lower quartiles are the ends of the box. Statistically significant differences between treatments are marked with an asterisk.

**Figure 3 animals-10-00401-f003:**
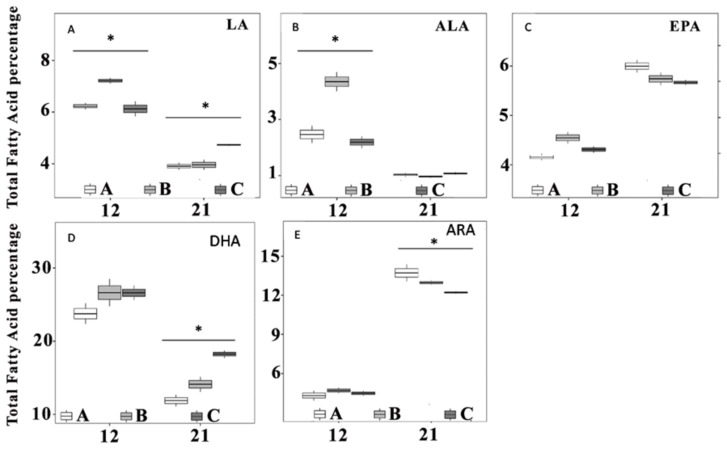
Larval Essential Fatty acids composition from the three treatments A (*Nannochloropsis*), B (*Chlorella*) and C (Spresso), after 12- and 21-days post hatch. (**A**) linoleic acid (LA), (**B**) alpha linoleic acid (ALA), (**C**) eicosapentonoic acid (EPA), (**D**) doxosahexaenoic acid (DHA), (**E**) arachidonic acid (ARA). Three samples 0.05 g of larvae per treatment were analyzed. Dots shown are the outer layers, whiskers indicate the maximum and minimum values excluding out layers, the line in the middle of box is the median value, and upper and lower quartiles are the ends of the box. Statistically significant differences between treatments are marked with an asterisk.

**Table 1 animals-10-00401-t001:** Total fatty acids percentage composition and mean (± Standard deviation) from the live feed supplied to the larvae for the three treatments.

Items	Live Feed Total FA % Used for Trial				
FA [%]	ROT (A)	ROT (B)	ROT (C)	ART (A)	ART (C)
C14:0	2.93 ± 0.62	1.87 ± 0.24	1.77 ± 0.11	2.60 ± 0.13	2.35 ± 0.15
C14:1	1.24 ± 0.12	0.64 ± 0.16	0.60 ± 0.03	1.09 ± 0.04	0.77 ± 0.03
C16:0	20.23 ± 0.10	18.51 ± 5.41	22.95 ± 0.26	16.03 ± 0.44	17.14 ± 0.50
C16:1	10.41 ± 0.23	2.04 ± 0.11	2.11 ± 0.18	16.01 ± 0.46	12.54 ± 0.37
C18:0	7.08 ± 0.14	4.07 ± 0.29	4.20 ± 0.02	5.22 ± 0.19	4.75 ± 0.10
C18:1n-9	4.89 ± 1.28	0.68 ± 0.07	0.86 ± 0.31	18.17 ± 0.27	17.06 ± 0.39
C18:1n-7	2.80 ± 0.11	2.38 ± 0.00	2.38 ± 0.17	13.05 ± 0.81	12.31 ± 0.22
C18:2n-6	13.97 ± 0.46	25.70 ± 2.06	24.19 ± 0.08	5.24 ± 2.65	5.28 ± 0.13
C18:3n-3	11.17 ± 0.42	36.42 ± 2.79	34.10 ± 0.90	2.10 ± 0.04	2.02 ± 0.25
C20:0	0.00 ± 0.00	0.00 ± 0.00	0.07 ± 0.10	0.08 ± 0.01	0.00 ± 0.00
C20:1n-9	1.12 ± 0.16	1.02 ± 0.03	0.86 ± 0.08	0.38 ± 0.05	0.36 ± 0.02
C20:4n-6	2.90 ±0.20	1.30 ± 0.16	1.03 ± 0.14	4.00 ± 0.28	3.70 ± 0.23
C20:3n-3	0.96 ± 0.08	1.89 ± 0.01	1.72 ± 0.11	0.03 ± 0.01	0.00 ± 0.00
C20:5n-3	10.73 ± 0.38	1.87 ± 0.02	1.72 ± 0.00	15.66 ± 1.12	15.20 ± 0.48
C22:0	1.08 ± 0.08	0.33 ± 0.07	0.30 ± 0.01	0.08 ± 0.02	0.32 ± 0.01
C22:1	0.22 ± 0.20	0.25 ± 0.11	0.07 ± 0.09	0.00 ± 0.00	0.00 ± 0.00
C22:5n-3	4.93 ± 0.16	1.05 ± 0.13	1.09 ± 0.04	0.10 ± 0.01	0.00 ± 0.00
C22:6n-3	2.09 ± 0.08	0.00 ± 0.00	6.50 ± 0.24	0.07 ± 0.02	6.21 ± 0.18
C24:1	0.09 ± 0.16	0.00 ± 0.00	0.00 ± 0.00	0.00 ± 0.00	0.00 ± 0.00
SFA	31.33 ± 0.62	24.77 ± 4.95	29.28 ± 0.30	24.09 ± 0.48	24.55 ± 0.76
MUFA	20.77 ± 1.43	7.01 ± 0.11	6.87 ± 0.67	48.71 ± 1.27	43.04 ± 0.25
PUFA	47.90 ± 1.95	68.22 ± 4.84	63.85 ± 0.97	27.21 ± 1.60	32.41 ± 0.51
n-3	29.88 ± 1.01	41.23 ± 2.63	38.63 ± 0.75	17.89 ± 1.09	23.43 ± 0.41
n-6	18.03 ± 0.95	26.99 ± 2.21	25.22 ± 0.22	9.31 ± 2.4	8.98 ± 0.10
n-3/n-6	1.66 ± 0.03	1.53 ± 0.03	1.53 ± 0.02	2.01 ± 0.5	2.61 ± 0.02

**Table 2 animals-10-00401-t002:** Larvae total fatty acids percentage composition and Standard deviation (±) from the three treatments (3 samples 0.05 g of larvae per treatment) after 12 and 21 days post hatching.

Items	Larval Total FA % at 12 dph			Larval Total FA % at 21 dph		
FA [%]	CON 12 (A)	CHLO 12 (B)	SP 12 (C)	CON 21 (A)	CHLO 21 (B)	SP 21 (C)
C14:0	0.89 ± 0.03	0.72 ± 0.06	0.56 ± 0.04	0.88 ± 0.20	0.74 ± 0.01	0.85 ± 0.04
C14:1	0.18 ± 0.01	0.13 ± 0.18	0.00 ± 0.00	0.33 ± 0.03	0.42 ± 0.06	0.36 ± 0.03
C16:0	23.44 ± 1.70	21.74 ± 1.49	22.01 ± 0.39	15.98 ± 0.23	16.16 ± 0.36	15.37 ± 0.24
C16:1	4.28 ± 0.07	3.53 ± 0.16	3.66 ± 0.27	5.99 ± 0.07	5.80 ± 0.00	6.08 ± 0.15
C18:0	9.77 ± 0.62	9.34 ± 0.63	9.84 ± 0.21	7.94 ± 0.03	7.65 ± 0.41	7.11 ± 0.06
C18:1n-9	12.75 ± 0.80	10.90 ± 0.02	11.96 ± 0.52	15.42 ± 0.01	15.51 ± 0.36	14.40 ± 0.17
C18:1n-7	3.46 ± 0.13	3.43 ± 0.23	3.55 ± 0.22	11.42 ± 0.31	11.10 ± 0.50	10.15 ± 0.57
C18:2n-6	6.23 ± 0.17	7.22 ± 0.14	6.12 ± 0.41	3.90 ± 0.20	3.96 ± 0.28	4.73 ± 0.05
C18:3n-3	2.47 ± 0.44	4.35 ± 0.49	2.19 ± 0.30	1.03 ± 0.09	0.96 ± 0.05	1.08 ± 0.06
C20:0	0.17 ± 0.08	0.00 ± 0.00	0.30 ± 0.02	0.12 ± 0.17	0.00 ± 0.00	0.40 ± 0.01
C20:1n-9	0.64 ± 0.45	0.46 ± 0.16	0.57 ± 0.06	0.78 ± 0.38	0.55 ± 0.11	0.25 ± 0.05
C20:4n-6	4.97 ± 0.04	4.55 ± 0.03	5.43 ± 0.24	6.00 ± 0.18	5.32 ± 0.18	5.25 ± 0.07
C20:3n-3	0.00 ± 0.00	0.00 ± 0.00	0.17 ± 0.01	0.00 ± 0.00	0.00 ± 0.00	0.00 ± 0.00
C20:5n-3	4.18 ± 0.53	4.57 ± 0.29	4.34 ± 0.27	13.72 ± 0.92	12.98 ± 0.24	12.22 ± 0.10
C22:0	0.47 ± 0.23	0.00 ± 0.00	0.10 ± 0.14	0.58 ± 0.28	0.89 ± 0.07	0.76 ± 0.05
C22:1	0.00 ± 0.00	0.00 ± 0.00	0.00 ± 0.00	0.04 ± 0.06	0.00 ± 0.00	0.00 ± 0.00
C22:5n-3	2.32 ± 0.21	2.43 ± 0.22	2.56 ± 0.23	4.31 ± 0.38	4.18 ± 0.45	3.12 ± 0.21
C22:6n-3	23.76 ± 1.98	26.63 ± 2.63	26.61 ± 1.42	11.54 ± 1.13	13.77 ± 1.42	17.88 ± 0.71
C24:1	0.00 ± 0.00	0.00 ± 0.00	0.00 ± 0.00	0.01 ± 0.01	0.00 ± 0.00	0.00 ± 0.00
SFA	34.75 ± 2.05	31.80 ± 2.07	32.82 ± 0.38	25.50 ± 0.11	25.44 ± 0.69	24.49 ± 0.38
MUFA	21.32 ± 1.32	18.45 ± 0.76	19.76 ± 1.08	34.00 ± 0.05	33.39 ± 0.18	31.25 ± 1.08
PUFA	43.93 ± 3.37	49.75 ± 2.83	47.43 ± 1.46	40.49 ± 0.06	41.17 ± 0.87	44.26 ± 1.46
n-3	32.73 ± 3.16	37.98 ± 2.65	35.87 ± 1.63	30.59 ± 0.08	31.89 ± 0.77	34.29 ± 1.63
n-6	11.20 ± 0.21	11.76 ± 0.18	11.55 ± 0.17	9.90 ± 0.02	9.27± 0.10	9.98 ± 0.17
n-3/n-6	2.92 ± 0.23	3.23 ± 0.18	3.11 ± 0.19	3.09 ± 0.01	3.44 ± 0.05	3.44 ± 0.19

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
