# Peer review of "Improvements on Live Feed Enrichments for Pikeperch (Sander lucioperca) Larval Culture"

_animals, 2020, doi:10.3390/ani10030401_

Round 1

Reviewer 1 Report

The manuscript describes a well conducted trial on a subject of interest for European pikeperch as well as for North American walley aquaculture research. The research group has the knowledge in larval pikeperch rearing, which is quite essential for reliable results. The methods have been adequately explained, but tank cleaning, lighting and possible removal of lipid layer on tank water surface need to be described. Please also describe if the parent fish were first time spawners or not. Statistical analyses is well described and performed. Please indicate hat is number of observations for Table 3 and Figure 3. I have not checked the references, not do I have comments on the English language, but it appears to be of good standard quality for scientific publications. I recommend accepting the manuscript after minor changes.

Author Response

Dear Reviewer,

Thank you for taking time to review our manuscript and for your comments and corrections, which has surely make this paper better.

Concerning your question about rearing tanks maintance and general conditions I have added the following information from line 98" Larvae were exposed to a 13/9 photoperiod and to dim light conditions (300 LUX) during the light hours. Daily siphoning was carried out to remove any debris from the bottom of the tanks. Order to reduce the lipid layer, surface skimmer we present in each tank."

About the broodstock question I added the answer in " line 83" Broodstock was spawned for the first time of the 2018 season but broodstock was first spawned on the previous year"

About the number of observation on Table 3 information sadded in line 308 "(3 samples 0.05 g of larvae per treatment) 

If you have any other comments or recommendations I will happy to address them.

Thank you

Carlos

Reviewer 2 Report

The paper entitled: Improvements on live feed enrichments for pikeperch (Sander lucioperca) larval culture”, describes the results of different enrichments treatments of live preys to use in the larval rearing of pikeperch.

The development of optimized feeding methods and protocols to increase the growth performance and survival of larvae is a priority in the culture of any species. For this reason, the results of this manuscript are of interest, and therefore, the authors should be well shown.

The authors have to write the manuscript according to the format of the journal. For example, the species name in italics, reference numbers of bibliographic citations, spaces between letters and signs, etc.

Line 26 and 27: the identification of treatments (a, b, c) is different in material and methods (Capital letters). The authors should homogenize the nomenclature of treatments.

Line 47 and 87, 89, 93…..: correct them and assign numbers

Line 74: remove parentheses

Line 94: parameters were daily monitored but Ammonia, nitrite and nitrate every 3 days. It is correct?

Line 103-104: species name, not only of the genus. Use abbreviation.

Table 1: DHP or DPH?. Perhaps table 1 gives little information and in order to save space, it could be translated into text.

Line 127: the authors mention “air bladder inflation”, but there are not results in the manuscript. They can delete it.

Line 131: numbers instead letters. The phrase “All larvae were ….. performance” is confuse

Line 180-181: the authors should define what is LA, ALA, etc. It's the first time they appear in the text.

Line 204-208: No reference can be made to figure 2 before figure 1.

Line 210: Why do authors use body weight?. It is not used in the rest of the manuscript. In general, the text confuses the reader with the continuous change of nomenclature

Figure 1: The font size is very small. They can barely see each other. 1c, the axes are not seen (colour). Treatments: Control, Chlorella and Spresso? They are different from the one used in the text. RNA/DNA or DNA/RNA?

Figure 1D, differences between 1 and 3 are difficult to see due to similar colours. Why treatment A at 16 dph is less than 100%?

Line 227: name of treatments with lower case?

Figure 2: the same as in the previous figure, very small letters, treatments nomenclature confuse.

Line 240-253: the treatment C is now called “sp”

Line 255-265: the name of the different fatty acids has not been previously described.

Table 2: the table title could be written better…. For example: Mean (± Standard Deviation). Why does the order of the treatments in the columns change? Treatments B, C and A instead of A, B, C. Delete line below C14:0, (and table 3).

Figure 3: missing legend in some figures. One more time treatments nomenclature is different to used in the text.

Line 303: Is reference 45 well put?

Discussion: In general, the authors continually change the nomenclature of the treatment instead of using treatment A, B and C. References or citations are incorrectly placed in the phrases or paragraphs.

Line 316: Pagrus major

Line 325: Scientific name of grouper?

Line 332: “…. and 37%, respectively”

Author Response

Dear Reviewer,

Thank you for taking time to review our manuscript and for your comments and corrections, which has surely make this paper better.

Thank you for addressing the mistakes on the editing side according to the journal format. I have gone through the whole manuscript and corrected all (you can see them all in red)

Concerning the homogenization of the treatment nomenclature across the manuscript, thank for pointing that out and I have changed it and use A, B and C, as the only term to name the treatments, hoping is more clear now. You can see all the changes in highlighted on red "A) Nannochloropsis occulata, B) Chlorella vulgaris, and C) a commercial enrichment - Selco, Spresso"

I have also changed all the legends in the figures so the term we used are A, B and C.

I have corrected the references on the suggested lines" current line 47:

"so that sufficient quantities of juveniles can be produced [11,12]."

Lines 88, 92 and 94:

"as previously cited by KÅ™ištan [33] and Blecha [31]. by Malinovskyi [29, 30],"

"until hatching occurred [30],"

I have also corrected few more from the Discussion and 

Parentheses removed from line 74.

I have clarified the timing for measurements: line 96 " Ammonia, Nitrite and nitrate levels were measured every three days. Ammonia (NH3 =0.21±0.05 mgL-1), nitrite (NO2 = 0.02±0.01 mgL-1) and nitrate (NO3 = 0.10±0.02 mgL-1). "

Added the species name line 106 " Artemia salina provided to larvae"

About Table 1, I have taken your recommendation and I have deleted it and added the information to the text: line 111-119 "Rotifers density was increased to 14 per ml from day 8 post hatching until day 12 post hatching. From day 13 post hatching rotifers density was progressively reduced, offering 10 rotifers per ml at day 13 and 8 rot/ml at day 14 post hatching. No more rotifers were added to the system after day 15 post hatching. Artemia feeding was applied in each experimental group from day 12 post hatching at a density of 2 artemia per ml. Prior to each feeding, residual counts were measured and feeding densities were steadily increased based on the counts. At day 13 and 14 post hatching density was increased to 3 and 4 per ml, respectively. On day 15 and 16 post hatching artemia density was 7 per ml and from day 17 to the end of the trial artemia density was 8 per ml. 

Air bladder was deleted from the text (it makes sense)

I have deleted the last part of sentence and leave as " samples were collected"

Fatty acids has been defined line 190: "rotifers used as preys (Linoleic Acid (LA) , Alpha linoleic acid (ALA),  Arachidonic acid (ARA), Eicosapentanoic acid (EPA) and Docosahexaenoic acid (DHA) as different response variables)"

I have changed the figures order so it correspond with the text in order or appearance

I have deleted the body weigh data, to avoid confusion.

Concerning the figures, I have changed all the figures fonts so they can be seen clearly, as well as homogenized the treatments term to A, B and C. The correct from is RNA/DNA Ration, I have corrected across the paper 

Figure 1 D I have changed the colour so the difference is clearly seen now, fix day 16 mistake (graphic mistake).

I have corrected and name all the treatments with upper case across the paper

Just like in figure 1 I have change the font size and homogenized the treatments terms

I have deleted "sp" and homogenized to "C"

Added fatty acids names description in line 190

I have rewritten the table description as advised "Table 2. Total fatty acids percentage composition and mean (± Standard deviation) from the live feed supplied to the larvae for the three treatments"

Also reorder the treatments so it reads A, B and C.

Added legends to figure 3 and homogenized the nomenclature.

Corrected reference 45 on the ref list "M. Fehér, E. Baranyai, E. Simon, P. Bársony, I. Sz, J. Posta, L. Stündl"

As mentioned before I have gone over the nomenclature and homogenized, and corrected mistakes with the references.

"Pagrus major" corrected

Scientific name of grouper: Epinephelus septemfasciatus

Added respectively

I hope I have been to address all your comments and corrections, thank you again for such a thorough revision.

Carlos

Round 2

Reviewer 2 Report

The authors have corrected the manuscript according to the recommendations.

Only the tables should be renumbered (and in the text).